# The Impact of Pedestrian Distraction on Safety Behaviours at Controlled and Uncontrolled Crossings

Amy O'Dell * , Andrew Morris , Ashleigh Filtness and Jo Barnes

Transport Safety Research Centre, School of Design and Creative Arts, Loughborough University, Loughborough LE11 3TU, UK
* Correspondence: a.odell@lboro.ac.uk

**Abstract:** To investigate differences in the safety behaviours of distracted and non-distracted pedestrians crossing roads, an unobtrusive observational study was conducted in Leicestershire, UK. Video recordings were taken of 1409 pedestrians crossing roads at controlled and uncontrolled crossing sites, both on a university campus and in urbanised town centre locations. On average, 42% of pedestrians were visibly distracted while crossing, and distracted pedestrians demonstrated significantly fewer safety behaviours than non-distracted pedestrians. They generally took longer to cross the road and made fewer looks towards the traffic environment, particularly at controlled crossings. Of all distraction activities, talking to another pedestrian had the most negative impact on safety behaviours. The findings highlight areas requiring further investigation, including distraction behaviours such as engaging with other pedestrians and supervising children. The results also identify that controlled crossings may benefit from targeted interventions to improve pedestrian safety.

**Keywords:** distraction; pedestrians; observations; road safety; vulnerable road user

## 1. Introduction

Pedestrian safety on roads is of great importance, as pedestrians lack the external protective measures that vehicles offer and are therefore more susceptible to injury in accidents. Globally, vulnerable road users contribute to more than half of all road user fatalities [1]. While efforts have led to a reduction in pedestrian casualties, recent UK statistics revealed an average of 115 serious pedestrian injuries per week between 2016 and 2021 [2]. One key factor that potentially contributes to road accidents is pedestrian distraction. Distraction, as defined in psychology, generally refers to stimuli or tasks that divert attention away from the primary task. In the context of pedestrian behaviour, distraction involves inattention while walking due to engaging in secondary activities, such as using mobile phones or conversing with others. This behaviour has been observed in a significant portion of pedestrians during both walking and crossing situations, and there is evidence to suggest that distracted pedestrians are more likely to be involved in accidents. An awareness of how distraction can impact safety and pedestrian behaviour is important when developing effective pedestrian distraction countermeasures and designing crossing infrastructure that can help prevent accidents.

### 1.1. Literature Review

The significance of pedestrian distraction in road safety has garnered attention across various studies, giving insight into the prevalence and effects of this behaviour. Research conducted in Australia found that approximately 20% of pedestrians were distracted due to smartphone usage [3], while self-report studies have indicated that 20% of pedestrians are at high risk of crossing while using a mobile phone, particularly among the 18–31 age group [4]. Further investigation, employing smartphone data, revealed that nearly half of participants' walking time was spent interacting with smartphones [5].

However, observed rates of distraction depend on methodological factors, location, demographics, and contextual variables. For instance, distraction prevalence was reported as low as 6% in Romania [6] but was significantly higher at 41% among pedestrians crossing roads on university campuses in the US [7]. A similar scenario emerged in US-based city observations, where 49% of pedestrians and drivers were found to be distracted [8].

Driver distraction, well-documented for its negative impact on road safety [9], has been paralleled by growing research on pedestrian distraction during walking and crossing. Experimental studies have consistently demonstrated connections between distraction and unsafe crossing behaviours. Virtual environment studies, for instance, have highlighted increased collision risk and reduced attention to the road environment during interactions with digital distractions [10,11]. Distracted pedestrians also exhibit delayed crossing initiation and slower walking speeds [12,13], along with heightened perceived workload and diminished situation awareness [14]. Moreover, various types of distractions have been associated with distinct behaviour patterns, with auditory-cognitive distractions leading to smaller gap acceptance and visual-manual distractions causing slower crossing [15].

Outdoor experiments further emphasised the impact of distraction, with texting while crossing significantly reducing attention to the road environment, followed by phone conversations and listening to music [16]. A meta-analysis consolidated findings from experimental and observational studies, indicating text messaging as the most detrimental distraction [17]. The consensus across experimental studies is that pedestrian distraction negatively impacts behaviour in controlled environments.

Real-world observation studies have revealed correlations between pedestrian distraction and adverse safety outcomes in diverse road settings. Distraction at city crossings has been linked to prolonged crossing times [18], increased incidence of critical events [3], and delayed responses to pedestrian signals [19]. Similarly, on university campuses, pedestrian distraction has been associated with increased attentional blindness [20] and reduced cautionary behaviours when crossing [21]. Nonetheless, there is evidence suggesting that distracted pedestrians might exhibit risk compensation behaviours by following traffic signals and using crosswalks more than non-distracted pedestrians [22].

Distinct distraction behaviours have also demonstrated varying impacts on gaze behaviour and overall safety. Listening to music, for example, was found to shift gazes toward the ground or straight ahead, potentially compromising awareness of traffic [23]. However, contradictory findings have emerged, with some studies indicating no significant differences in safety behaviour between headphone use and non-distraction [24]. Similarly, while one study highlighted mobile gaming as the riskiest behaviour, followed by web-browsing [25], another found that talking on a mobile phone had the greatest impact on safety, followed by texting/browsing [26]. This incongruence highlights a lack of consensus on the most impairing distraction activities in naturalistic road settings.

Mediating factors influencing distracted pedestrian behaviour have also been identified. High pedestrian traffic has been linked to altered behaviour among mobile phone-using pedestrians, resulting in increased near-miss occurrences and reduced crossing speeds [27]. Additionally, sleep deprivation amplifies the safety risk for distracted pedestrians [28]. Gender-based disparities have also been found, with women across various age groups being more prone to distraction while walking than their male counterparts [29,30]. This could be because women are more likely to be carrying their personal items, are more likely to multitask, and are more afraid of harassment, and so may engage in distractions to appear busy and avoid confrontation.

## 1.2. Research Questions

Considering the significance of pedestrian distraction and its potential impacts on road safety, this study aims to investigate observable safety behaviours during controlled and uncontrolled road crossings through video-recorded observations. Safety behaviours examined encompass crossing time, crossing initiation time, and visual behaviour. This study sought to provide insights into the implications of distraction for pedestrian safety

at road crossings by addressing the following research questions: Are there discernible differences in the observed safety behaviours of distracted and non-distracted pedestrians at road crossings? Do patterns of safety behaviours vary based on the type of distraction? Does the type of crossing influence safety behaviours?

By exploring these research questions, this study aims to enhance our understanding of the role of pedestrian distraction in road safety, and its implications for behaviour in real-world settings.

## 2. Method

This study utilised direct video-recorded observations as an unobtrusive method to collect naturalistic pedestrian behaviour data. Video recorded observations have been validated as a reliable method for analysing pedestrian-vehicle interactions, particularly when attempting to capture timing measurements and behavioural sequences [31]. The observations were conducted discretely, so as not to influence the behaviour of pedestrians entering the filming area and to ensure that the data reflected typical pedestrian activities. Ethics approval was granted by Loughborough University.

### 2.1. Site Locations

A preliminary investigation was conducted to select the most suitable locations for carrying out observations in Leicestershire, UK. This included analysing vehicle–pedestrian incident data from 2014–2018 to identify accident hotspots [32] and assessing the local area to identify controlled and uncontrolled crossing points. Controlled crossings are those which give legal priority to pedestrians, whereas uncontrolled crossings do not give legal priority to pedestrians. The controlled crossings of interest are zebra crossings (indicated by painted white stripes on the roadway) and traffic light-controlled crossings (indicated by pedestrian-operated traffic lights). Uncontrolled crossings are generally informal places to cross, where pedestrians may take cues from the environment, such as dropped kerbs. Six observation sites were chosen, representing three types of crossing (zebra, traffic light-controlled, uncontrolled) previously identified as high risk for distracted pedestrians [33]. Sites were chosen both on Loughborough University campus and in urbanised town centre areas of Loughborough and Leicester. Site characteristics can be seen in Table 1, and a map of the crossing locations can be found in Supplementary Materials Figure S1.

**Table 1.** Observation site characteristics (traffic flow calculated as vehicles per hour during the observation window).

| Site | Location | Carriageway | Crossing Type | Crossing Location | Road Speed | Traffic Flow |
|------|----------|-------------|---------------|-------------------|------------|--------------|
| 1 | Campus | Dual | Zebra | Midblock | 15 mph | 150 |
| 2 | Campus | Dual | Traffic light | Midblock | 40 mph | 1566 |
| 3 | Campus | Dual | Uncontrolled | Junction | 15 mph | 213 |
| 4 | Town | Dual | Zebra | Midblock | 30 mph | 354 |
| 5 | Town | Single | Traffic light | Junction | 30 mph | 246 |
| 6 | Town | Single | Uncontrolled | Midblock | 30 mph | 138 |

Two of the campus crossing locations were inside the campus environment (Site 1 and Site 3), where the speed limit of vehicles is restricted to 15 mph and only used by authorised vehicles (e.g., staff, students, visitors, public buses, and delivery vehicles). The other campus crossing (Site 2) was on a main public road separating two university department buildings. The town crossing locations were all in public areas of high pedestrian footfall, near to local amenities. Two were in Loughborough town centre (Site 5 and Site 6), and one was in Leicester city centre (Site 4). The site in Leicester was chosen due to the lack of zebra crossings in Loughborough with comparable traffic and pedestrian flow.

### 2.2. Procedure

A GoPro 8 (1080p video resolution, 30 fps) camera was used to record footage by being securely attached to crossing infrastructure or placed on a nearby power box providing a long shot full view of the crossing at eye level from one side (see Figure 1 as an example). Videos were recorded at each site for one hour, providing a total of 6 h. Footage was collected between November 2021 and April 2022 during daylight hours (between 9 a.m. and 2 p.m.) with clear weather conditions on weekdays and weekends.

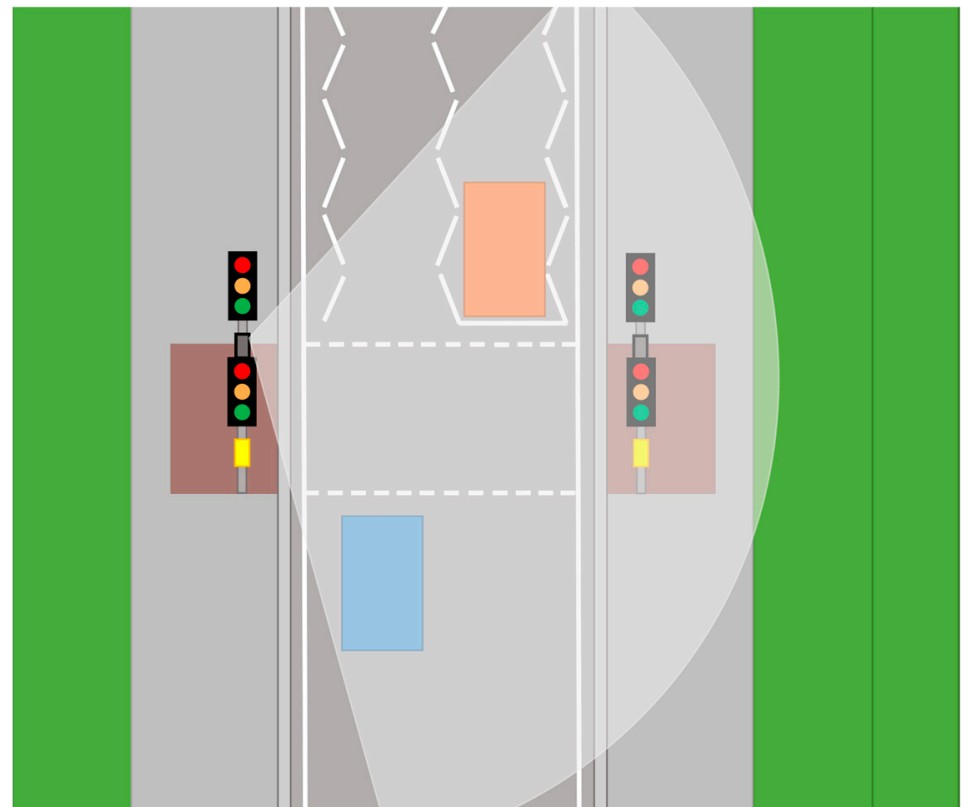

**Figure 1.** Example of crossing location with shaded area indicating filming zone.

### 2.3. Observation Criteria

Observation criteria were created in line with previous work [3,18]. Safety behaviours were chosen as they are either linked with reduced awareness for the road environment (e.g., looking before/during crossing the road) or increase the time spent on the road (e.g., crossing time). The pedestrian features, environmental features, and safety behaviours that were collected can be seen in Table 2.

**Table 2.** Observation criteria (pedestrian/environmental features and safety behaviours).

| Observation Criteria | Additional Information |
|---|---|
| Pedestrian features | |
| Gender | Female/male |
| Estimated age category | 18–30, 31–60, 61+. Any pedestrians who appeared to be under the age of 18 were not included at the data coding stage |
| Presence of other pedestrians | Were other pedestrians present during the crossing interaction? This included pedestrians under 18 years old |

**Table 2.** *Cont.*

| Observation Criteria | Additional Information |
|---|---|
| Group size | Groups were defined as multiple pedestrians observably engaging in a social interaction such as talking or holding hands. Pedestrians under 18 years old were included in group size, but infants in pushchairs or being carried by adults were not |
| Distraction behaviours before and during crossing | Whether pedestrians were engaging in observable distractions and distraction behaviour type. Examples include using headphones, talking to another pedestrian, and texting/browsing on a mobile phone |
| Encumbered | Whether pedestrians were holding/pushing anything while crossing; for example, mobile phones, shopping bags, infants, pushchairs, and wheelchairs |
| Environmental features | |
| Traffic dynamics | Was a vehicle present during the crossing? Was the vehicle stationary/moving? Did the pedestrian wait until there was no traffic before crossing? |
| Signal status | Whether pedestrians followed the green/red signals at traffic light-controlled crossings |
| Safety behaviours | |
| Visual behaviour before and during crossing | Number of looks made to the left/right by visibly turning the head and whether pedestrians looked one way or both ways |
| Crossing bounds | Whether pedestrians started and ended their crossing within the marked boundaries of the crossing area. This was not applicable to uncontrolled crossings |
| Initiation duration | Time taken between approaching the crossing and stepping from the pavement onto the road |
| Crossing time | Time taken from placing the first foot off the curb to first foot on the pavement on the other side of the road |
| Crossing speed | The width of the crossing was divided by the time taken to cross, giving the speed in m/s. Crossing width was obtained using an augmented reality ruler phone app |
| Critical events | "An observable event which would end in an accident unless one of the involved parties slows down, changes direction or accelerates to avoid a collision"—Risser 1985 |

### 2.4. Data Coding and Satistical Analyses

All videos were coded by one researcher. The first five minutes of each video were coded by a second researcher and codes were compared for reliability. Coding was consistent between researchers. Each row represented one pedestrian making a complete crossing from one side of the road to the other. Each pedestrian was only counted once, even if they crossed the road multiple times.

Data were input into a Microsoft Office Excel spreadsheet and analysed using IBM SPSS v.27 statistical software (IBM Corp, Armonk, NY, USA). An alpha level of 0.05 was used to determine statistical significance (*).

A series of independent samples t-tests were conducted to compare the influence of distraction (two levels: distraction by any cause and no distraction) on the safety behaviours of crossing time, crossing initiation time, and looks to the left/right before and during crossing, within each of the six sites. One-way ANCOVAS were used to analyse differences in safety behaviours (three levels: crossing speed, crossing time, and number of looks before and during crossing) based on distraction type (four levels: talking to another pedestrian, using headphones, mobile phone browsing, and supervising children) at controlled and uncontrolled crossings.

## 3. Results

### 3.1. Overall Data

A total of 1409 pedestrians were observed across the six sites. Table 3 shows the demographic data of pedestrians observed at each site, along with distraction prevalence and crossing in the presence of other pedestrians.

**Table 3.** Pedestrian demographics by percentage at each site (gender, age, distraction prevalence, crossing in the presence of other pedestrians, crossing in a social group).

| | Site Location | | | | | |
|---|---|---|---|---|---|---|
| | **1**<br>**Campus Zebra** | **2**<br>**Campus Traffic Light** | **3**<br>**Campus Uncontrolled** | **4**<br>**Town Zebra** | **5**<br>**Town Traffic Light** | **6**<br>**Town Uncontrolled** |
| **Gender %** | | | | | | |
| Female | 30 | 55.1 | 38.9 | 49.5 | 50.9 | 54.9 |
| Male | 70 | 44.9 | 61.1 | 50.5 | 49.1 | 45.1 |
| **Age %** | | | | | | |
| 18–30 | 82.5 | 93.9 | 95 | 22.2 | 33.6 | 28.2 |
| 31–60 | 16.6 | 5.7 | 5 | 51.4 | 49.9 | 53.1 |
| 61+ | 0.9 | 0.4 | 0 | 26.4 | 16.5 | 18.8 |
| **Distraction %** | | | | | | |
| Before | 45.3 | 68.4 | 40 | 31.5 | 32.5 | 33.8 |
| During | 48.4 | 57.8 | 44 | 32.4 | 32.5 | 35.7 |
| **Other pedestrians %** | 59.2 | 95.1 | 47 | 54.2 | 78.2 | 47.4 |
| **Groups %** | 43.9 | 41.4 | 28.3 | 26.4 | 63.5 | 36.6 |

Chi-square tests were performed to examine the relationship between gender and distraction. Women were more likely than men to cross while distracted at Site 3 only ($\chi^2(1, 113) = 6.44$, $p = 0.01$).

Chi-square tests were also performed to examine the relationship between age and distraction. Pedestrians in the 18–30 age group were more likely to cross while distracted than the older age (61+) groups at Site 1 ($\chi^2(2, 222) = 8.32$, $p = 0.02$), Site 4 ($\chi^2(2, 216) = 25.31$, $p < 0.001$), Site 5 ($\chi^2(2, 381) = 20.94$, $p < 0.001$), and Site 6 ($\chi^2(2, 213) = 18.55$, $p < 0.001$).

Distraction prevalence during crossing ranged from 32–58%. After averaging across the six sites, approximately 42% of pedestrians were observably distracted during road crossing. A more detailed breakdown of distraction types can be seen in Figure 2.

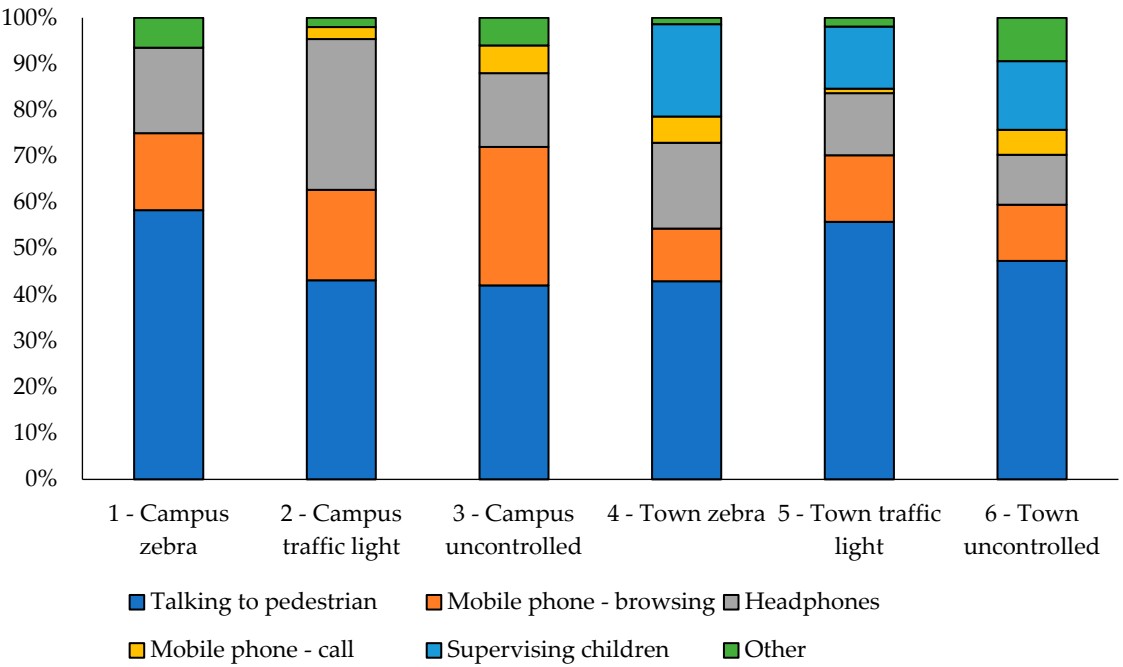

**Figure 2.** Percentage of observable distraction behaviours by crossing site.

Across the entire dataset, the most frequent form of distraction was talking to another pedestrian (48.23%), followed by using headphones (18.34%) and browsing on a mobile phone (17.39%). However, in town centre locations (Site 4), supervising children was the third most frequent form of distraction (16.12%).

*3.2. Distraction and Safety Behaviours*

3.2.1. Crossing Time and Crossing Initiation

Independent samples *t*-tests were conducted for each of the site locations to investigate the impact of distraction on crossing time (s) and crossing initiation (s). The means and standard deviations across the whole dataset can be seen in Table 4.

**Table 4.** Mean and standard deviation crossing time (s) and crossing initiation time (s) for distracted (D) and non-distracted (ND) pedestrians across the six sites.

|  | 1—Campus Zebra | | 2—Campus Traffic Light | | 3—Campus Uncontrolled | | 4—Town Zebra | | 5—Town Traffic Light | | 6—Town Uncontrolled | |
|---|---|---|---|---|---|---|---|---|---|---|---|---|
|  | D | ND | D | ND | D | ND | D | ND | D | ND | D | ND |
| Crossing time (s) | 4.62 (1.27) | 4.17 (1.28) | 7.36 (1.00) | 7.05 (1.09) | 6.42 (1.07) | 6.72 (1.72) | 5.55 (0.89) | 5.22 (0.96) | 4.83 (0.82) | 4.74 (0.78) | 7.52 (1.14) | 7.47 (1.92) |
| Crossing initiation (s) | 0.80 (0.54) | 0.95 (0.71) | 2.05 (2.28) | 1.70 (1.84) | 0.62 (0.53) | 0.91 (0.55) | 1.31 (0.70) | 1.09 (0.51) | 0.94 (0.61) | 0.92 (0.73) | 0.95 (0.59) | 0.92 (0.58) |

For crossing time, it was found that distracted pedestrians took significantly longer to cross the road than non-distracted pedestrians at Site 1, the campus zebra crossing (t(212) = 2.56, *p* = 0.01, Cohen's d = 0.35), as well as Site 2, the campus traffic light-controlled crossing (t(231) = 2.27, *p* = 0.02, Cohen's d = 0.30), and Site 4, the town zebra crossing (t(202) = 2.38, *p* = 0.02, Cohen's d = 0.36).

For crossing initiation, distracted pedestrians (M = 0.91, SD = 0.55) had significantly shorter crossing initiation times than non-distracted pedestrians (M = 0.62, SD = 0.53) at Site 3, the campus uncontrolled crossing (t(87) = 2.43, *p* = 0.02, Cohen's d = 0.54).

### 3.2.2. Visual Behaviour

Independent samples t-tests were conducted for each of the site locations to investigate the impact of distraction on the number of looks made to the left and right both before and during crossing, as well as the total number of looks made during the entire crossing interaction. The mean and standard error for each crossing type can be seen in Figures 3–5.

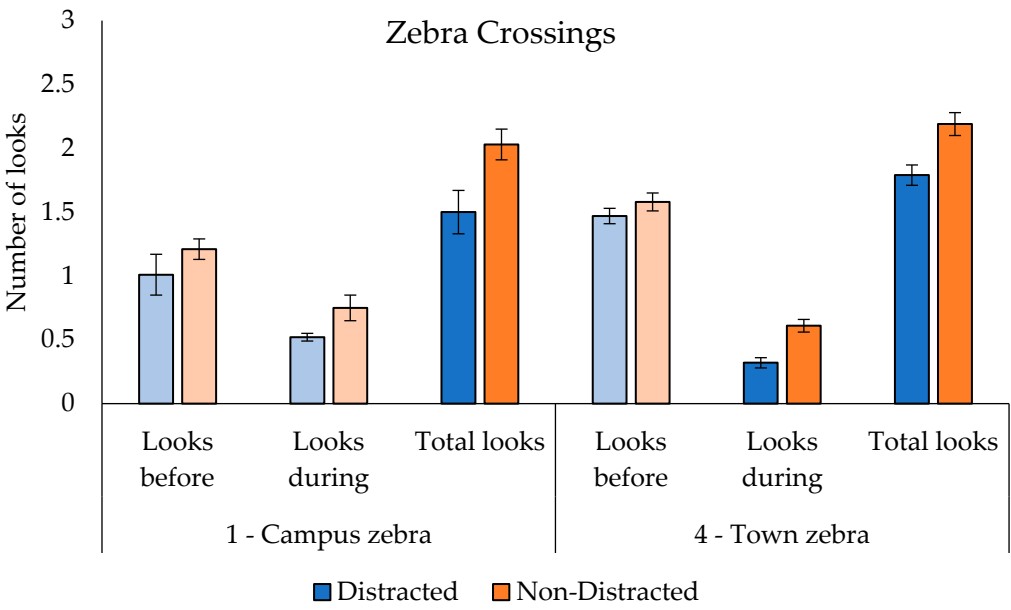

**Figure 3.** Mean number of looks made before, during, and in total for distracted and non-distracted pedestrians at zebra crossing sites. Error bars represent one standard error of the mean. Dark shaded bars indicate a statistically significant difference between distracted and non-distracted groups.

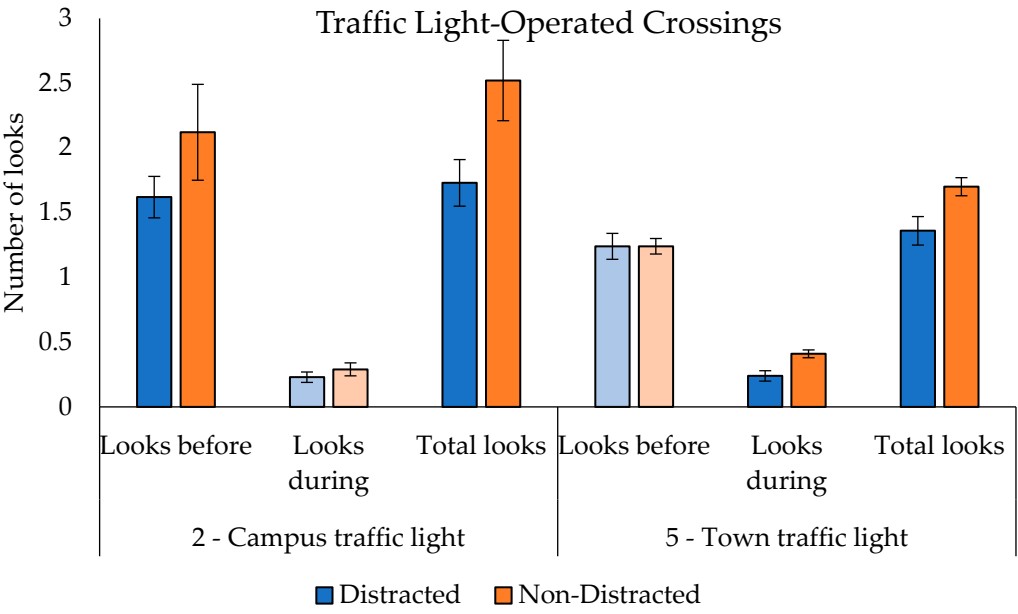

**Figure 4.** Mean number of looks made before, during, and in total for distracted and non-distracted pedestrians at traffic light-controlled crossing sites. Error bars represent one standard error of the mean. Dark shaded bars indicate a statistically significant difference between distracted and non-distracted groups.

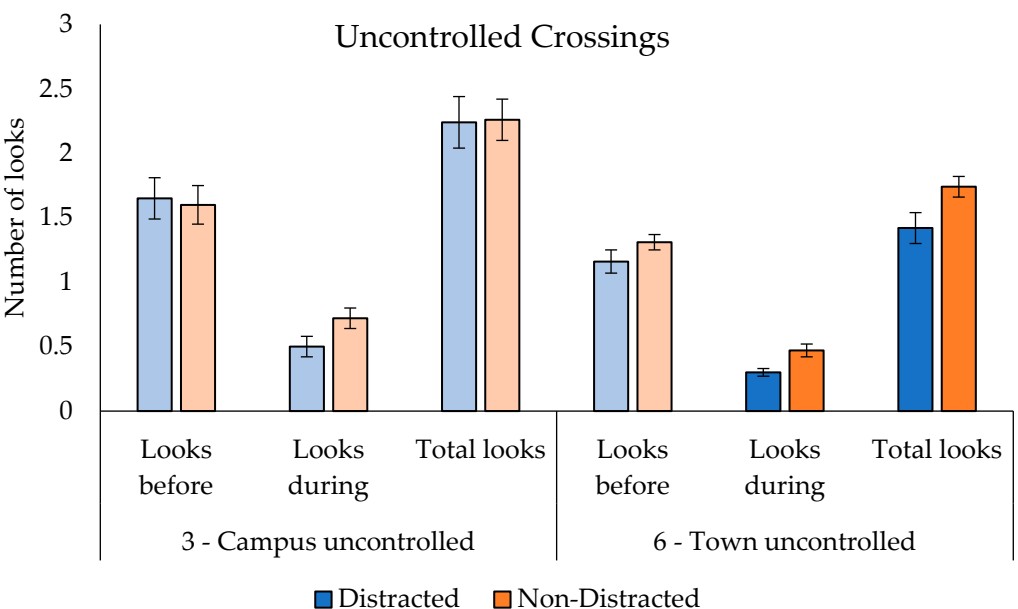

**Figure 5.** Mean number of looks made before, during, and in total for distracted and non-distracted pedestrians at uncontrolled crossing sites. Error bars represent one standard error of the mean. Dark shaded bars indicate a statistically significant difference between distracted and non-distracted groups.

In terms of looks made before crossing, distracted pedestrians (M = 1.62, SD = 1.98) made significantly fewer looks to the left/right before crossing than non-distracted pedestrians (M = 2.12, SD = 3.03) at Site 2, the campus traffic light-controlled crossing (t(224) = 1.46, *p* = 0.03, Cohen's d = 0.20).

For looks made during crossing, distracted pedestrians (M = 0.32, SD = 0.53) made significantly fewer looks during crossing than non-distracted pedestrians (M = 0.61, SD = 0.63) at Site 4, the town zebra crossing (t(156.46) = 9.29, *p* < 0.01, Cohen's d = 0.50). Distracted pedestrians (M = 0.24, SD = 0.43) also made significantly fewer looks during crossing than non-distracted pedestrians (M = 0.41, SD = 0.57) at Site 5, the town traffic light-controlled crossing (t(374) = 2.79, *p* < 0.01, Cohen's d = 0.34). Finally, distracted pedestrians (M = 0.30, SD = 0.52) made significantly fewer looks during crossing than non-distracted pedestrians (M = 0.47, SD = 0.61) at Site 6, the town uncontrolled crossing (t(172.10) = 2.03, *p* = 0.04, Cohen's d = 0.31).

For total looks made during the crossing interaction, only Site 3, the campus unmarked crossing, had a non-significant difference between distracted and non-distracted pedestrians. At all other sites, distracted pedestrians made significantly fewer total looks than non-distracted pedestrians, as seen in Table 5.

**Table 5.** Results from independent t-test analyses for total number of looks by distracted and non-distracted pedestrians at the six crossing sites. * indicates significant difference between distracted and non-distracted groups.

|  | Distracted | | Not Distracted | | | | | |
|---|---|---|---|---|---|---|---|---|
|  | *M* | *SD* | *M* | *SD* | *df* | *t* | *p* | Cohen's *d* |
| 1—Campus zebra * | 1.5 | 1.28 | 2.03 | 1.27 | 204 | 2.97 | <0.01 | 0.42 |
| 2—Campus traffic light * | 1.73 | 2.05 | 2.52 | 2.9 | 219 | 2.37 | 0.02 | 0.31 |
| 3—Campus uncontrolled | 2.24 | 2.17 | 2.26 | 1.12 | 89 | 0.06 | 0.95 | 0.01 |
| 4—Town zebra * | 1.79 | 1.06 | 2.19 | 1.06 | 198 | 2.55 | 0.01 | 0.38 |
| 5—Town traffic light * | 1.36 | 1.07 | 1.7 | 1.2 | 365 | 2.43 | 0.02 | 0.30 |
| 6—Town uncontrolled * | 1.42 | 1.05 | 1.74 | 0.97 | 211 | 2.26 | 0.03 | 0.32 |

*3.3. Impact of Distraction Type*

One-way ANCOVAs were conducted for controlled and uncontrolled crossings across the entire dataset to examine the effects of distraction type on the safety behaviours of crossing speed (m/s) and total number of looks, controlling for age and gender. Crossing speed was used as a comparison as it is a standardised measure of risk exposure across all sites.

There was a statistically significant effect of distraction type on crossing speed at controlled crossings ($F(5, 402) = 4.60$, $p < 0.001$, partial $\eta^2 = 0.05$). The covariates of age and gender had no significant effect. Bonferroni post hoc tests revealed that crossing speed was significantly faster when using headphones compared to browsing on a mobile phone ($p = 0.04$) or talking to another pedestrian ($p < 0.001$).

There was no statistically significant effect of distraction type on crossing speed at uncontrolled crossings ($F(5, 100) = 0.92$, $p = 0.47$, partial $\eta^2 = 0.04$). The covariates of age and gender also had no significant effect.

There was a statistically significant effect of distraction type on the total number of looks made before/during crossing at both controlled crossings ($F(5, 395) = 5.42$, $p < 0.001$, partial $\eta^2 = 0.06$) and uncontrolled crossings ($F(5, 103) = 2.52$, $p = 0.03$, partial $\eta^2 = 0.11$). The covariates of age and gender had no significant effect at either crossing type. Bonferroni post hoc tests revealed that the total number of looks was significantly higher when using headphones compared to talking to another pedestrian at controlled ($p < 0.001$) and uncontrolled crossings ($p = 0.02$).

## 4. Discussion

This study found that between 32 and 58% of pedestrians were distracted when crossing the road, both in a university campus setting and urbanised town environment, which is in line with previous findings [5,7,33]. The most popular form of distraction was talking to another pedestrian, followed by browsing on a mobile phone and using headphones. It is important to note that although mobile phone distractions have received the most amount of research attention, they do not appear to be the most commonly occurring distraction activities. Furthermore, in town centre locations, supervising children was the third most frequent form of distraction. This should be an area of specific investigation, as it is not clear if supervising children makes pedestrians more distracted or more focused on the road to protect the child. It is important to consider whether the system allows for safe completion of this complex crossing task.

The first research question addressed by this study was to investigate differences in the observed safety behaviours of distracted and non-distracted pedestrians at road crossings. The results confirm that distracted pedestrians demonstrate fewer safety behaviours than non-distracted pedestrians, similar to previous naturalistic observation research [3,7,18,21]. In this study, distracted pedestrians made significantly fewer looks toward traffic at five of the six sites, suggesting reduced visual attention toward the road environment both before and during crossing. Furthermore, distracted pedestrians were also slower to cross at five of the six sites, and significantly slower at both campus and town zebra crossings as well as the campus traffic light-controlled crossing. Slower crossing increases the time spent on-road, and therefore the opportunity to be involved in a conflict interaction with traffic. Combined with reduced visual attention, distracted pedestrians are not only paying less attention to the road, but their exposure to risk is also greater.

The second research question related to whether patterns of safety behaviours vary based on distraction type. The findings demonstrate that talking to another pedestrian had the most significant negative impact on safety behaviours and was associated with slower crossing speeds at controlled crossings, as well as fewer looks toward the road environment at both controlled and uncontrolled crossing types. Talking to another pedestrian requires cognitive, auditory, and visual processing [34], which may explain why it impacts pedestrians to a greater extent than technological distractions, which often only involve one or two modalities. It is recommended that talking to another pedestrian should be

the focus of further study, as only a small amount of research (e.g., [21]) has considered the impact of this distraction type, and so this behaviour is still not very well understood within the context of road crossing.

Furthermore, of all distraction activities, using headphones had the least negative impact on safety behaviours. This may have implications for interactions between pedestrians and electric vehicles. Electric vehicle noise output is much lower than that of internal combustion engines, with research being conducted to predict the intrusive noise of electric vehicles [35] in order to further reduce it. However, although previous research has found that distracted pedestrians are poorer at detecting approaching vehicle noise, their gap acceptance is unaffected [36]. The results from the present study also demonstrate that pedestrians who are using headphones, and therefore less exposed to auditory cues, show more observable safety behaviours than pedestrians engaging in other forms of distraction. This may suggest that engine noise is not an essential indicator for safe crossing.

The final research question concerned safety behaviours at different crossing types. Generally, there were fewer observed differences in safety behaviours between distracted and non-distracted pedestrians at uncontrolled crossing types compared to controlled crossing sites. This may be due to the different rules surrounding uncontrolled crossings; pedestrians are aware that vehicles are not required to give way to them, and therefore may demonstrate additional caution even while distracted by a competing activity. However, crashes do still occur at controlled crossings [31] despite pedestrians having right of way, so distracted pedestrians may be more surprised by the presence and behaviours of vehicle drivers, and therefore at greater risk of conflict at these crossing sites.

## 5. Conclusions

This research study investigated pedestrian distraction behaviours and their impact on safety at road crossings. The study was conducted in both a university campus setting and an urban town environment. Between 32 and 58% of pedestrians were found to be distracted when crossing the road, with common distractions including talking to other pedestrians, browsing on mobile phones, and using headphones.

Distracted pedestrians demonstrated fewer safety behaviours when crossing roads than non-distracted pedestrians. They exhibited reduced visual attention towards traffic by making fewer looks before and during crossing, and were slower to cross, increasing their exposure to potential conflict with vehicles.

The impact of distraction types on safety behaviours was also explored. Conversations with other pedestrians had the most significant negative impact on safety behaviours, leading to slower crossing speeds and reduced attention to the road environment. Using headphones had the least negative impact on safety behaviours, suggesting that pedestrians using headphones might compensate for reduced auditory cues by being more cautious visually.

This study also considered safety behaviours at different crossing types. There were fewer differences in safety behaviours between distracted and non-distracted pedestrians at uncontrolled crossings compared to controlled crossings. This could be due to pedestrians exercising more caution at uncontrolled crossings, where vehicles are not obligated to yield.

Based on the conclusions drawn from the research study, several pedestrian distraction countermeasures and interventions can be considered as best practice to enhance road safety by aiming to mitigate the negative effects of distraction. One of these is to enhance the visibility of road crossings with clear signage, pavement markings, and warning signs on the ground to remind pedestrians to focus on crossing safely. Another suggestion is to develop technological solutions, such as smartphone apps that provide real-time notifications to pedestrians approaching road crossings. These notifications could be auditory, visual, or tactile, and would remind users to put down their devices and pay attention while crossing. Additionally, though there are safety issues among distracted pedestrians, the main burden of responsibility should still be placed on vehicle drivers, who must remain vigilant to pedestrian behaviour. Therefore, distraction awareness

should be incorporated into driver education, with the aim of fostering better understanding and empathy among drivers and encouraging them to be more cautious around distracted pedestrians.

As well as strengths, this study also had some limitations. Pedestrians were observed under naturalistic conditions at a variety of different crossing sites. However, due to the unobtrusive observation methodology, participant demographics such as age were estimated and are therefore subject to error. Furthermore, although video recordings increase the accuracy and detail of the data, it can still be difficult to identify precise behaviours, such as eye gaze patterns. To address this limitation, more cameras could be installed at different angles on sites to distinguish behaviours more easily. Additionally, as the data were collected in Leicestershire (UK), the results may not be generalisable to other locations, such as larger cities or other countries with different road rules and traffic culture.

In light of this study's findings, there are several avenues for future research that could enhance understanding of pedestrian behaviour. Investigation into how supervising children affects pedestrian behaviour presents an intriguing avenue for further exploration, as the impact of this behaviour on road safety remains unclear. It is crucial to evaluate whether the existing system accommodates safe completion of this intricate crossing task. Additionally, talking to another pedestrian emerged as the most detrimental distraction behaviour, affecting crossing speeds and visual attention across both controlled and uncontrolled crossings. This unique form of distraction requires cognitive, auditory, and visual processing, possibly explaining its heightened impact compared to technological distractions that involve fewer sensory modalities. Given the relative scarcity of research focusing on pedestrian-pedestrian interactions, further exploration is recommended to investigate its specific implications for road crossing safety. Further research could also consider the role of pedestrian intention and decision-making during road crossing, which cannot be captured through unobtrusive observation. Additionally, the differences in safety behaviours at different crossing types indicate the need for development and testing of context-specific interventions to reduce the risks associated with pedestrian distraction.

**Supplementary Materials:** The following supporting information can be downloaded at: https://www.mdpi.com/article/10.3390/futuretransp3040065/s1, Figure S1: Location map of crossing observation sites.

**Author Contributions:** Conceptualization, A.O., A.M. and A.F.; methodology, A.O., A.M., A.F. and J.B.; software, A.O.; validation, A.O., A.M., A.F. and J.B.; formal analysis, A.O.; investigation, A.O.; resources, A.M.; data curation, A.O.; writing—original draft preparation, A.O.; writing—review and editing, A.M., A.F. and J.B.; visualization, A.O.; supervision, A.M., A.F. and J.B.; project administration, A.O. and A.M.; funding acquisition, A.M. All authors have read and agreed to the published version of the manuscript.

**Funding:** This research was funded by the International Research Centre to Study the Effects of Autonomous Vehicles on Vulnerable Road-Users, a Research England International Investment Initiative (I3). Funding number 131910.

**Institutional Review Board Statement:** The study was approved by the Ethics Committee of Loughborough University (protocol code 6036, 18 January 22).

**Informed Consent Statement:** Not applicable.

**Data Availability Statement:** The data presented in this study are available on request from the corresponding author. The data are not publicly available due to GDPR and ethics implications.

**Conflicts of Interest:** The authors declare no conflict of interest. The funders had no role in the design of the study; in the collection, analyses, or interpretation of the data; in the writing of the manuscript; or in the decision to publish the results.

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
