# Peer review of "The Impact of Pedestrian Distraction on Safety Behaviours at Controlled and Uncontrolled Crossings"

_futuretransp, doi:10.3390/futuretransp3040065_

Round 1

Reviewer 1 Report

This paper determined the “The impact of pedestrian distraction on safety behaviours at controlled and uncontrolled crossings”. This paper aims to investigate the differences in the safety behaviours of distracted and non-distracted pedestrians crossing roads. An unobtrusive observational study was conducted in the UK for 1409 people. The results indicated that, on average, 42% of pedestrians were visibly distracted while crossing and distracted pedestrians demonstrated significantly fewer safety behaviours than non-distracted pedestrians. The contributions can be applied to improve traffic safety. This paper is relevant and valuable to the readers of Future transportation. However, several places need to be revised. There, I recommend publication of this paper subject after major revision. The following changes to improve this paper was suggested.

  1. The article needs more literature review and summarizes the related work of predecessors. As we can see from the whole article, the reader needs help understanding the current situation of the development of green operation and management of the port, nor can it reflect the innovation of the work done by the author.
  2. The literature needs to be more sufficient. The literature section needs to thoroughly discuss the theory and concepts that build the proposed model. You need to discuss the basic theory, keyword, and your constructs. Precision in the application of theory to create the proposed model needs improvement. I suggest that you separate your important title and discuss it systematically.
  3. The methodology is appropriate for the size and level of data in question. The sample selected to be surveyed was considered carefully before the results were drawn. How to choose these samples? Why is this survey in the UK? Can the results be employed in other regions?
  4. The suggestion for further research needs to be stronger. Please give some valuable recommendations for further study.

Non.

Reviewer 2 Report

An interesting paper to investigate the impacts of different behaviours on traffic safety from pedestrains who are distracted or non-distracted under the controlled and uncontrolled traffic scences. The detail analysis is implemented based on the data collected from the university campus and town centre. The follwoing suggestions may be useful and supportive for the publication of paper:

1) Most readers will focus on the conclusions which they don't know or are not familiar with. Therefore, what the new results from the analysis found in the paper should be addressed.

2) The secondery conclusions which readers have the basic concerns but are comfuse should be described in detail.

3) Some conclusions which could not lead to the definite results concerned with the traffic safety events should be explained simply with some suggestions to avoid the bad cases.

4) Concerning the limitation from the video recordings, being difficult to identify precise behaviours, if the research and analysis will continue, more cameras are suggested to be installed on the different angles on sites to distinguish the behaviours easily.

Reviewer 3 Report

My general comment is as follows. 

1. Interesting topic.

2. Well written.

3. Well cited. 

My specific comments is as follows. 

4. Introduction. Section is too long. Focus introduction section on importance of topic and research questions. Create a separate literature section which includes all of the relevant citations.

5. Introduction. Eliminate numbers in parentheses before research questions.

6.  Method. Site Locations. Insert map of site locations.

7. Method. Procedure. Insert graphic to depict camera locations relative to different crossings.

8. Method. Observation Criteria. Pedestrian features, environmental features and safety behaviours content belongs in a table.

9.  Results. Overall data. Figure 1. Add crossing site labels to horizontal axis.

10. Discussion. Paragraph 6 and paragraph 7 belong in the conclusions section.

11. Conclusions. Elaborate on the pedestrian distraction countermeasures and interventions that constitute best practices based on the results.

12. Conclusions. Glaring omission not to elaborate on trajectory of future research based on the results.

Minor grammatical errors.

Round 2

Reviewer 3 Report

Well done.